# Involvement of miR-190b in Xbp1 mRNA Splicing upon Tocotrienol Treatment

**DOI:** 10.3390/molecules26010163

**Published:** 2020-12-31

**Authors:** Roberto Ambra, Sonia Manca, Guido Leoni, Barbara Guantario, Raffaella Canali, Raffaella Comitato

**Affiliations:** Council for Agricultural Research and Economics—Research Centre for Food and Nutrition, via Ardeatina 546, 00178 Rome, Italy; roberto.ambra@crea.gov.it (R.A.); sonia.manca84@gmail.com (S.M.); guido.leoni@gmail.com (G.L.); barbara.guantario@crea.gov.it (B.G.); raffaella.canali@crea.gov.it (R.C.)

**Keywords:** endoplasmic reticulum stress, miR-190b, apoptosis, tocotrienol

## Abstract

We previously demonstrated that apoptosis induced by tocotrienols (γ and δT3) in HeLa cells is preceded by Ca^2+^ release from the endoplasmic reticulum. This event is eventually followed by the induction of specific calcium-dependent signals, leading to the expression and activation of the gene encoding for the IRE1α protein and, in turn, to the alternative splicing of the pro-apoptotic protein sXbp1 and other molecules involved in the unfolded protein response, the core pathway coping with EndoR stress. Here, we showed that treatment with T3s induces the expression of a specific set of miRNAs in HeLa cells. Data interrogation based on the intersection of this set of miRNAs with a set of genes previously differentially expressed after γT3 treatment provided a few miRNA candidates to be the effectors of EndoR-stress-induced apoptosis. To identify the best candidate to act as the effector of the Xbp1-mediated apoptotic response to γT3, we performed in silico analysis based on the evaluation of the highest ∆ in Gibbs energy of different mRNA–miRNA–Argonaute (AGO) protein complexes. The involvement of the best candidate identified in silico, miR-190b, in Xbp1 splicing was confirmed in vitro using T3-treated cells pre-incubated with the specific miRNA inhibitor, providing a preliminary indication of its role as an effector of EndoR-stress-induced apoptosis.

## 1. Introduction

Endoplasmic reticulum (EndoR) stress occurs when the capacity of the EndoR to fold proteins becomes saturated. The excess of unfolded/misfolded proteins in the lumen stimulates the release of the chaperone protein immunoglobulin heavy-chain binding protein (BiP), activating and redirecting different unfolded protein response (UPR) signaling pathways [1], to correct the protein-misfolding stress. Regulators of UPR include transmembrane proteins, such as PKR-like endoplasmic reticulum kinase (PERK), activating transcription factor (ATF6), and inositol-requiring enzyme (IRE1α), which triggers a conformational shift that activates its endoribonuclease domain. This nuclease then catalyzes a unique cytoplasmic mRNA-splicing reaction, specifically cleaving out 26 nucleotides from X-box binding protein 1 (Xbp1) mRNA [2,3]. Because of a reading frame shift, translation of the cleaved mRNA yields the highly active transcription factor Xbp1, which upregulates multiple foldases, oxidoreductases, and intracellular trafficking components [4], and induces pro-inflammatory cytokine production, lipid and hexosamine biosynthesis, the hypoxia response [5], and apoptosis [6,7].

We previously demonstrated that induction of apoptosis in HeLa cells by γ- and δ-tocotrienols (T3s) is associated with EndoR stress and the UPR through Ca^2+^ release and Xbp1–mRNA alternative splicing [8]. T3s, belonging to the vitamin E family, are often considered to have minor action compared to tocopherols [9,10] but have relevant functions related to cell-type-dependent growth arrest and apoptosis [10,11,12]. Importantly, recent data indicate that EndoR stress is also regulated by micro-ribonucleic acids (miRNAs) [13,14,15]. miRNAs are a class of small, noncoding RNAs involved in post-transcriptional gene expression regulation through sequence-specific activation of translational repression or mRNA de-adenylation and degradation mediated by the Argonaute (AGO) RNA-induced silencing complex (RISC) [16]. Ameyar-Zazoua et al. demonstrated that AGO1 proteins are involved in alternative splicing [17]. Several miRNAs are shown to respond to different nutritionally related molecules and components in different types of tissues [18]. For instance, in vivo vitamin E deficiency induces differential miRNA expression in rat liver tissues [19].

The present study aimed to identify miRNAs as upstream regulators of the UPR in EndoR stress, using HeLa cells exposed to T3s as a model. Our results indicate a direct involvement of miR-190b in Xbp1 splicing and in the antiproliferative effect of γT3 and δT3.

## 2. Results

### 2.1. miRNA Profiling and Validation of miRNA Expression

We evaluated the effect of γT3 treatment of HeLa cells on a panel of 671 miRNAs using TaqMan Array MicroRNA Cards, as reported in the Material and Methods. The expression of 298 miRNAs was technically not achievable because the Ct of one or more technical replicates was undetermined. Of the remaining 373 miRNAs, 328 were excluded according to a biological significance fold-change threshold set arbitrarily to one (log_2_ ratio) for treated versus control cells, whereas 45 miRNAs were considered (of which 11 were downregulated and 34 upregulated and equally distributed between the two cards) (Table 1). The number of significantly modulated miRNAs was further reduced to 10 upregulated and only 1 downregulated miRNA (in bold in Table 1) using a Limma parametric test combined with the Bonferroni method for false discovery adjusted with a *p*-value threshold of < 0.05. To validate results obtained through TaqMan Cards, γT3 incubation of HeLa cells was repeated and modulation of a set of miRNAs was remeasured using miRNA-specific TaqMan probes and the human pseudogene RNU6B as a calibrator [20]. This strategy confirmed that three miRNAs, namely miR-190b (*p*-value = 0.05), miRNA-215 (*p*-value = 0.0009), and miRNA-148a (*p*-value = 0.001), were significantly upregulated by γT3.

### 2.2. Identification of Target Genes Modulated by miR-190b

AGO is an essential component of the RNA-induced silencing complex (RISC) and plays a central role in recognizing miRNA targets and therefore in the modulation of mRNA translation processes [21]. Therefore, to identify candidate effectors for the Xbp1 response to γT3, we performed in silico analysis and identified the highest ∆ in Gibbs binding energy of different AGO protein complexes (Figure 1A) between miRNAs and previously identified γT3-regulated mRNAs in HeLa cells [8]. Compared to all the other AGO complexes considered, the one involving miR-190b and Xbp1–mRNA displayed the highest ∆ binding energy, suggesting miR-190b as the best candidate to modulate Xbp1 expression (Figure 1A,B).

### 2.3. Effects of Modulation of miR-190b on bp1 mRNA

As shown in our past works, either γT3 or δT3 induces the EndoR stress pathway by specifically activating the IRE1α pathway of the UPR [8]; therefore, we studied the effect of miR-190b on Xbp1, the transcription factor regulated by the IRE1α pathway, using both forms of tocotrienols.

Firstly, we evaluated the effects of T3s on Xbp1 mRNA in pre-miR-190b (pre190b)- or anti-miR-190b (an190b)-pretreated cells. None of these treatments significantly affected the level of Xbp1 mRNA (Appendix A), suggesting that the effect of miR-190b could be related to other regulations, possibly the ability to positively modulate mRNA Xbp1 splicing, increasing the level of the truncated form of Xbp1.

We previously reported that γT3 and δT3 induced the alternative splicing of Xbp1 mRNA in HeLa cells [20]. Using an190b-induced miR-190b inhibition, here we showed that miR-190b is responsible for Xbp1 splicing (Figure 2). Pre-incubation with an190b prevented the activation of the active form of Xbp1 (sXbp1) in all treatments in a dose-independent way (Figure 2A). However, even if pre190b-induced stimulation of miR-190b alone did not influence Xbp1 splicing, it increased T3-induced splicing in the presence of both γT3 at the highest concentration (20 µg/mL) and δT3 at lower concentrations (5 and 10 µg/mL) (Figure 2B), underlining a greater effect of δT3 compared to γT3. At a higher concentration of δT3 (20 µg/mL), we observed many dead cells, so we could not collect RNAs and proteins. Notably, we did not observe an inhibitory effect of an190b on Xbp1 splicing when cells were treated with two other well-known triggers of EndoR stress, brefeldin A (BFA) and tunicamycin (TUN), instead of T3s (data not shown). These results indicated that miR-190b is responsible for the T3-induced splicing of Xbp1, possibly leading to the formation of the small, pro-apoptotic form, sXbp1.

### 2.4. miR-190b Modulates T3 Antiproliferative Effects

We assessed whether miR-190b inhibition could recover the viability of HeLa cells treated with T3s. As shown by EVOS imaging (Figure 3), treatment with γT3 or δT3 induced cell death, as demonstrated by cellular shrinking and detachment in the culture dish in comparison to control cells. Such effects were not observed when cells were co-treated with an190b, suggesting a role of miRNA in the execution of cell death induced by T3s in HeLa cells. However, an190b or pre190b alone had no significant effect on the cell number.

The effect of γT3 or δT3 administration in combination with pre190b or an190b on HeLa cell viability was also assessed by analyzing the cell proliferation rate using 5-bromo-2′-deoxyuridine (BrdU). Figure 4 shows that γT3 (20 µg/mL) or δT3 (10 µg/mL) reduced the number of proliferating HeLa cells by about one-half with respect to control cells and that pretreatment with pre190b quantitatively doubled tocotrienols’ antiproliferative effect. Conversely, an190b pretreatment restored the proliferation of δT3-treated HeLa cells compared to control cells. A similar trend was observed for γT3; however, the effect was lower and statistically not significant. Finally, the administration of either pre- or an190b alone had no effect on cell proliferation.

## 3. Discussion

Various stress factors, such as hypoxia, starvation, oxidative insults, changes in pH, Ca^2+^ depletion, hypoglycemia, ATP depletion, and viral infections, can disturb EndoR homeostasis and cause a perturbation of protein-folding mechanisms [22], resulting in unfolded protein accumulation, known as EndoR stress. In response to EndoR stress, cells activate the UPR signaling pathway to overwhelm stress and re-establish homeostasis; conversely, unresolved severe stress can lead to the activation of apoptotic pathways.

One of the three distinct but overlapping signaling branches of the mammalian UPR involves the activation of IRE1α, a dual-function type I transmembrane protein with Ser/Thr protein kinase and endoribonuclease (RNase) activities. The latter, which becomes active upon IRE1α dissociation from BiP, dimerization, and trans-autophosphorylation, is essential for the execution of the UPR through splicing of the Xbp1 mRNA, its only characterized target to date [23]. Xbp1 mRNA undergoes unconventional (cytoplasmic, spliceosome-independent) splicing to generate a potent basic leucine zipper domain (bZIP)-containing transcription factor [2,24,25]. Removal of 26 nucleotides from the mammalian Xbp1 mRNA results in a translational frame switch, encoding a protein containing 376 amino acids compared to the 261-amino-acid protein encoded by the unspliced mRNA. Both forms of Xbp1 can bind the EndoR stress element; however, sXbp1 activates the UPR far more potently than its unspliced form [2,25]. The unspliced protein (uXbp1) is unstable in the cell and can heterodimerize with ATF6 and sXbp1 to promote their proteasomal degradation [25]. The sXbp1 protein also upregulates the expression of proteins involved in EndoR protein folding, endoplasmic-reticulum-associated degradation (ERAD), and vesicular trafficking [26,27].

In the present study, we observed that γT3 induced a significant modulation of the expression of a specific set of miRNAs, using miRNA cards. A subset of miRNAs confirmed by means of RT-PCR was submitted to an in silico study based on the evaluation of the ∆G resulting from the molecular interaction of each miRNA with the *Xbp1*–Argonaute complex. This approach identified miR-190b as the best candidate to target Xbp1, and the involvement of the miRNA in the execution of T3-induced apoptosis in HeLa cells was demonstrated through the inhibition of its expression using its specific anti-miR.

Different miRNAs play an important regulatory role in EndoR stress signaling [13,14,15,28,29]. The observed increase in a specific miRNA moiety agrees with some previous published observations. However, the reported outcomes are different. Byrd et al. reported that miR-30c-2-3p* is upregulated in embryonic fibroblasts treated with tunicamycin, targeting the 3′-UTR of Xbp1 mRNA, thereby affecting the survival of cells undergoing ER stress [30]. Conversely, miR-214 has been reported to repress Xbp1 expression in hepatocellular carcinoma cells [31]. The same authors observed that miR-214, miR-199a-3p, and miR-199a-5p levels significantly reduced in the majority of examined hepatocarcinomas and in other liver cancer lines (HepG2 and SMMC-7721) in comparison with their nontumor counterparts [31].

Using exogenous inhibition (with anti-miR) or stimulation (with pre-miR) of miR-190b, here we demonstrated that the Xbp1-mediated antiproliferative effects of T3s depend specifically on miR-190b function. While miR-190b inhibition abolished T3s-induced Xbp1 splicing and cellular effects, exogenous stimulation of the miRNA had the opposite effect in only T3-treated cells. Accordingly, inhibition of miR-190b did not abolish BFA- or TUN-induced Xbp1 splicing (data not shown) or miR-190b expression (Appendix A). Yoshida et al. demonstrated that IRE1α mediates the unconventional splicing of Xbp1 mRNA [2]. In our previous work, we showed that T3s specifically induce IRE1α phosphorylation [8]. Based on here-reported new preliminary data, we hypothesize that after migrating to the cytoplasm, miR-190b interacts with the IRE1α–uXbp1 complex and drives uXbp1 degradation, redirecting Xbp1 splicing to the active form of the transcription factor (sXbp1).

Even though the role of the miRNA pathway in mediating a stress reaction is well-recognized, the functions of specific miRNAs in regulating particular aspects of cellular-stress-responsive mechanisms are just beginning to emerge.

Our data highlight a novel role of a miRNA, i.e., miR-190b, in gene modulation by regulation of alternative splicing in the UPR system, even though further experiments are needed for understanding and analyzing the molecular basis of this modulation.

## 4. Materials and Methods

### 4.1. Chemicals

Purified T3 was provided by Dr. Hiroyuki Yoshimura of the Eisai Food and Chemical Co., Ltd. (Tokyo, Japan). The purity was ~99% for all T3s. The actual concentration of T3 solutions was determined spectrophotometrically from the specific extinction coefficients (ε_296_ γT3 = 90.5) before each experiment. Stock solutions of T3 were stored at −20 °C in aliquots and diluted to the desired concentration in dimethyl sulfoxide (DMSO).

### 4.2. Cell Lines and Treatments

HeLa cells were obtained from the American Tissue Culture Collection (Manassas, VA, USA). The cells were grown in DMEM (Euroclone, Pero, Milan, Italy) supplemented with 10% fetal bovine serum (FBS; Sigma-Aldrich, St. Louis, MO, USA), 100 U/mL penicillin, 100 µg/mL streptomycin (Pen/Strep, Euroclone), 2 mM glutamine (Euroclone), and 1% non-essential amino acids (Sigma-Aldrich, St. Louis, MO, USA). The cells were maintained at 37 °C in a humidified atmosphere of 5% CO_2_.

Before any experimental session, the cells were synchronized in G_1_/G_0_ by starvation in serum-free medium for two days. Once synchronized, 300,000 cells were seeded onto multiwell plates. Purified T3s were dissolved in DMSO and individually added to the culture medium. When not differently indicated in the text, the incubation time was 24 h. Purified γT3 (20 µg/mL) and δT3 (10 µg/mL) were added to the medium. The concentrations of T3s used in this experimental set had previously been tested in HeLa cells [21]. Control cells (CC) were treated with an equal volume of DMSO alone.

Tunicamycin and brefeldin A were used as inducers of EndoR stress. Preliminary investigations indicated that they have identical effects on our cellular model. Treatments with 2.5 µg/mL of BFA for 8 h or 5 µg/mL of TUN for 4 h were therefore used as positive controls for EndoR stress. The cells were observed under an EVOS microscope.

### 4.3. TaqMan miRNA Array Profiling

Total RNA was extracted using an miRVana miRNA Isolation Kit (Thermo Fisher Scientific, Waltham, MA, USA) using 600 µL lysis/binding solution and eluting with 100 µL of nuclease-free water following the manufacturer’s protocol. The concentration and quality of the extracted RNA were evaluated using Nanodrop 1000 (Thermo Fisher Scientific, Waltham, MA, USA). miRNA expression profiling was performed using TaqMan Array MicroRNA Cards A and B (Thermo Fisher Scientific) on an ABI Prism 7900HT Real-time System (Applied Biosystems, Foster City, CA, USA), which allowed us to evaluate 671 unique assays specific to human miRNAs covering Sanger miRBase v10, including candidate endogenous control assays (MammU6, RNU6B, RNU24, RNU43, RNU44, and RNU48). A total of 300 ng RNA was reverse-transcribed using Megaplex RT primers, prior to pre-amplification following the manufacturer’s protocol, in a GeneAmp PCR System 9700 thermocycler (Applied Biosystems). Raw cycle threshold (Ct) values were determined using ABI Prism Sequence Detection software (SDS version 2.4; Applied Biosystems). The raw Ct value represents the value where the amount of amplified cDNA reaches a defined threshold. Data normalization was carried out using a global mean normalization method [32] and the Bonferroni method was used to adjust for a false discovery rate.

### 4.4. Identification of Gene Targets

The list of modulated miRNAs was intersected using RmiR package, with the differential gene expression profile obtained by comparing microarray experiments of HeLa cells treated with γT3 versus untreated cells [8]. RmiR allows the search in public prediction databases for miRNA target pairs extracted from differential expression profiles. All the modulated miRNA target pairs annotated in MIRANDA (PMID: 20799968) [33] and microcosm (PMID: 17991681) [34] were extracted using RmiR package (10.18129/B9.bioc.RmiR.Hs.miRNA) and further filtered using PITA (PMID: 17893677) [35] by retaining only 10 pairs with a predicted target score of ≤10 ΔG. The surviving pairs were prioritized using MiREN software (PMID: 26825463) [36], which allows identification of the miRNA target nucleotide strands that are most likely to be complexed with a favorable decrease in Gibbs free energy (ΔG) within the AGO protein-binding site.

### 4.5. Real-Time qPCR Validation of Differentially Expressed miRNAs

miRNAs profiling was performed using TaqMan Array MicroRNA Cards (Thermo Fisher Scientific, Waltham, MA, USA), as described above. Selected miRNAs among the most significantly differentially expressed ones were validated by real-time reactions using an Applied Biosystems 7500 Fast Real-time PCR system equipped with individual TaqMan probes. The synthesis of cDNA from 40 ng of total RNA was carried out in a 15 μL reaction volume using the TaqMan MicroRNA Reverse Transcription kit (Applied Biosystems) following the manufacturer’s instructions. The reaction contained three different 5X RT primers (miR-190b, miR-215, and miR-148a) and the RNU6B 60X RT primer as a calibrator. The reverse transcription conditions were as follows: 16 °C for 30 min, then 42 °C for 30 min, and, finally, 85 °C for 5 min.

For each sample, the relative quantity (RQ) was determined according to the 2^−ΔΔCT^ method and the expression of RNU6B for normalization.

### 4.6. Transfections with miRNA Mimics and Inhibitors

To clarify the involvement of miR-190b in EndoR stress induced by tocotrienol treatment, HeLa cells were transfected with pre-miR-190b and anti-miR-190b (Thermo Fisher Scientific, Waltham, MA, USA) using lipofectamine RNAiMax (Invitrogen, Carlsbad, CA, USA) following the manufacturer’s instructions. The cells were then treated with γT3 or δT3 for 24 h and the RNA harvested using Trizol (Invitrogen). Anti-miRNA and pre-miRNA are chemically modified single-stranded nucleic acids designed to specifically bind to and inhibit endogenous miRNA and mimic the effects of endogenous mature miRNA molecules, respectively, enabling the investigation of miRNA’s biological effects by silencing or upregulating its own expression. We observed that the administration of anti-miR-190b (an190b) was associated with a significant decrease in γT3- and δT3-induced miR-190b expression. Conversely, the presence of pre-miR-190b (pre190b) was associated with increased levels of the messenger (Appendix A).

### 4.7. Alternative Splicing Xbp1 mRNA

cDNA was synthesized from total RNA using One-Step RT-PCR SuperScript III reverse transcriptase (Invitrogen) according to the manufacturer’s protocol. The primers used for PCR (F: AAACAGAGCAGCAGTCCAGACTGC; R: TCCTTCTGGGTAGACCTCTGGGAG) were specific for human sXbp1 (hsXbp1). The PCR conditions were as follows: 95 °C for 5 min, 95 °C for 1 min, 58 °C for 30 s, 72 °C for 30 s, and 72 °C for 5 min, with 35 cycles of amplification. A 488 bp amplicon was generated from unspliced Xbp1 (uXbp1). PCR amplicons were digested by PstI. The PstI cleavage site is located within the 26 nt intron of uXbp1, which allows differentiation between the uXbp1 amplicon (cut PCR product) and sXbp1. After digestion, only PCR amplicons derived by uXbp1 showed two fragments (a 263 bp and a 225 bp amplicon). Digested and not-digested PCR products were resolved on 2% agarose gels, stained with EtBr. The PCR fragments were visualized with the UVIpro Bronze Imaging System (UVitec, Cambridge, UK).

### 4.8. Proliferation Assay

Proliferation was obtained by incorporation of 5-bromo-2′-deoxyuridine (BrdU). Cells were seeded in 96-well plates (4000 cells/well) and pulse-labeled with BrdU 24 h before measurement. BrdU incorporation was measured using the DELFIA Cell Proliferation Kit (PerkinElmer Life Sciences, Waltham, MA, USA) according to the manufacturer’s protocol. Fluorescence was measured using a fluorescence detector (Infinite® 200 PRO; TECAN, Männedorf, Switzerland).

### 4.9. Statistical Analysis and Data Presentation

Statistical analysis was performed using R software from the R Foundation for Statistical Computing (Vienna, Austria). Data were analyzed by one-way ANOVA with repeated measures, followed by Tukey’s test; for non-normal but homogeneous data, we used the Kruskal–Wallis test. *p*-values of ≤0.05 were considered statistically significant.

Figures show one out of at least three independent experiments providing similar results or the mean (±S.E.) of at least three experiments.

## Figures and Tables

**Figure 1 molecules-26-00163-f001:**
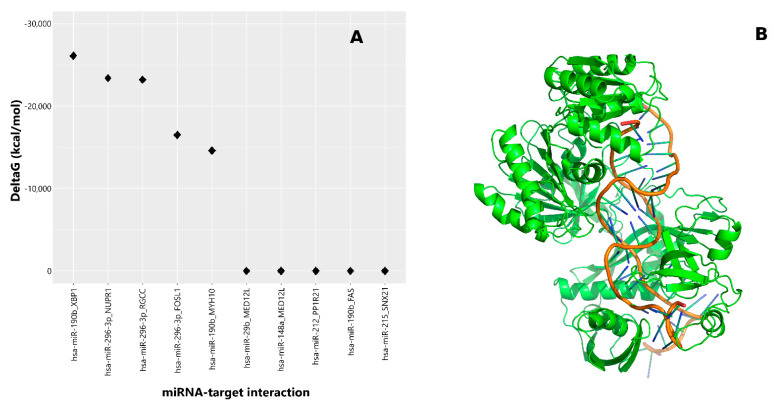
In silico analysis for the identification of favorable miRNA–mRNA target complexes induced in HeLa cells treated with γT3: (**A**) Gibbs free energy decrease was estimated using MiREN software, testing for the 10 most favorable predicted miRNA–mRNA target pairs. The best complex was found in the hsa-miR-190b–Xbp1 pair. (**B**) Tridimensional model of the ternary complex formed by miR-190b, Xbp1 mRNA, and the T.hermus thermophilus AGO protein (PDB code: 3F73), which is considered an appropriate structural model representative of the eukaryotic members of AGO proteins.

**Figure 2 molecules-26-00163-f002:**
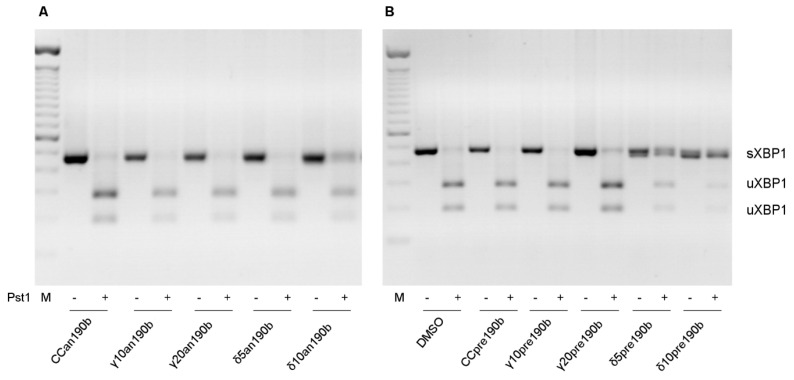
Xbp1–mRNA alternative splicing. Effects of γT3 (10 or 20 µg/mL for 24 h) and δT3 (5 or 10 µg/mL for 24 h) on HeLa cells transfected with anti-miR-190b (Panel (**A**), an190b) and pre-miR-190b (Panel (**B**), pre190b) and on mRNA alternative splicing of Xbp1. CC, control cells. PCR amplicons were digested with PstI. Digested and not digested PCR products were resolved on 2% agarose gels, stained with EtBr. The figure shows 1 of at least 3 representative experiments.

**Figure 3 molecules-26-00163-f003:**
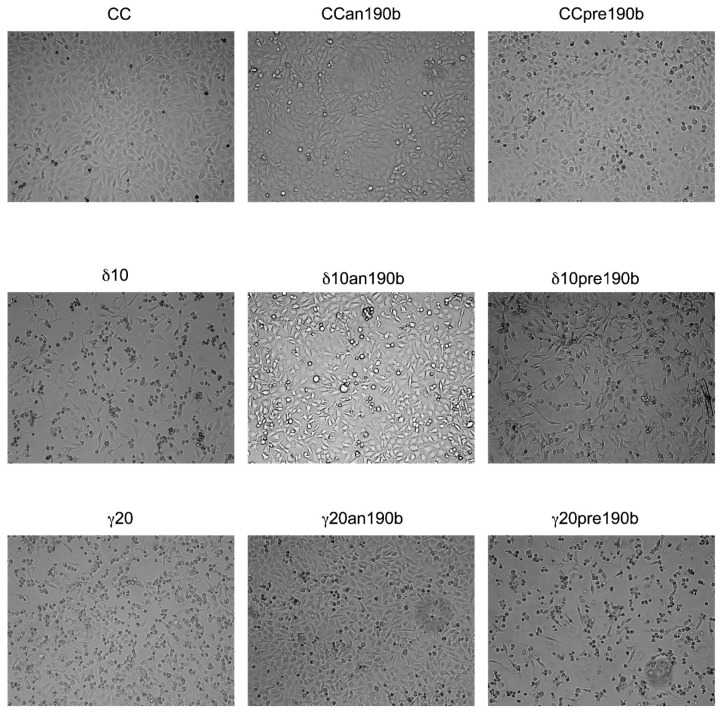
Effect of anti-miR-190b (an190b) and pre-miR-190b (pre190b) on cell proliferation after treatment with γT3 (20 µg/mL for 12 h) and δT3 (10 µg/mL for 12 h) in HeLa cells. CC, control cells. The figure (10× enlargement) shows 1 of at least 3 representative experiments.

**Figure 4 molecules-26-00163-f004:**
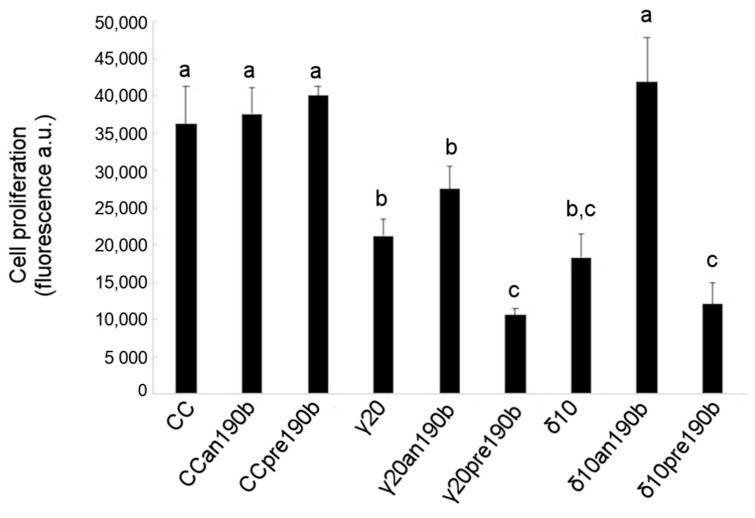
Effects of γT3 (20 µg/mL for 24 h) and δT3 (10 µg/mL for 24 h) on the proliferation of HeLa cells transfected with pre-miR-190b (pre190b) and anti-miR-190b (an190b). Results are reported as the mean ± SD of data obtained from 5 independent experiments and expressed as arbitrary fluorescence values normalized per 10,000 cells. Bars show a log_2_ fold-change (treated vs. control, CC). Data were analyzed by one-way ANOVA with repeated measures, followed by Tukey’s test. Different letters (a, b, c) indicate significant differences (*p* < 0.05).

**Table 1 molecules-26-00163-t001:** RQ (log base 2) of miRNAs in γT3-treated HeLa cells (*n* = 2) with respect to control cells (*n* = 4). Data are expressed as the mean value. miRNAs appearing as bold type have a *p*-value threshold of <0.05 according to a Limma test combined with the Bonferroni method for a false discovery rate; * they refer to the specific probe used in the miRNA PCR array.

Downregulated		Upregulated	
miR-372	−3.97	**miR-215**	**3.66**
miR-10a *	−3.29	**miR-148a**	**3.18**
**miR-26a-1 ***	**−1.95**	**miR-29b-1 ***	**2.73**
miR-522	−1.51	**miR-190b**	**2.72**
miR-329	−1.49	miR-517a	2.69
miR-27b *	−1.38	**miR-25 ***	**2.69**
miR-125a-3p	−1.27	**miR-616 ***	**2.34**
miR-551b *	−1.27	**miR-296-3p**	**2.21**
miR-369-3p	−1.14	miR-517c	1.86
miR-769-5p	−1.11	**miR-886-5p**	**1.69**
miR-565	−1.01	miR-132 *	1.67
		miR-409-5p	1.56
		miR-632	1.49
		**miR-320**	**1.47**
		**miR-212**	**1.42**
		miR-801	1.38
		miR-9 *	1.33
		miR-29b	1.31
		miR-424 *	1.31
		miR-132	1.24
		miR-106b *	1.19
		miR-219-1-3p	1.16
		miR-645	1.16
		miR-516a-3p	1.13
		miR-624 *	1.12
		miR-768-3p	1.1
		miR-200a	1.09
		miR-29a *	1.08
		let-7c	1.07
		miR-29a	1.07
		miR-374b *	1.06
		miR-184	1.05
		miR-638	1.01
		miR-221 *	1.01

## Data Availability

Data and the MIREN software for the in silico experiments are available from Guido Leoni.

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
