# Peer review of "Involvement of miR-190b in Xbp1 mRNA Splicing upon Tocotrienol Treatment"

_molecules, 2020, doi:10.3390/molecules26010163_

Round 1

Reviewer 1 Report

This is a resubmitted manuscript that I reviewed once previously. After reading the revised version I found that the authors properly addressed my concerns and revised the manuscript appropriately. I have no more concerns before accepting the manuscript.

Author Response

we thank the reviewer for the rapid revision.

No further revisions were required

Reviewer 2 Report

In this manuscript, Ambra et al. indicate that the treatment with T3 induces the expression of a specific set of miRNAs in HeLa cells. Among them,  miR-190b seems to be important since it was demonstrated that this miRNA is responsible for the T3-induced  splicing of Xbp1 and that miR-190b inhibition could recover the  viability in of HeLa cells treated with T3.

It has been demonstrated before that different miRNAs play an important regulatory role in EndoR stress signaling. These studies contribute to the overall knowledge about this process.

The experiments presented in the paper are properly designed and analyzed. The conclusions sound interesting.

Minor points:

Introduction, line 46-47: it would be more convenient to present UPR regulators in order of their action in the cell during this response: first PERK, then ATF6, and IRE1.

Introduction, line 59, this part of the sentence “T3s are vitamins E with different action and function in respect to  tocopherols” is not very clear. Please, add more information about tocotrienols and tocopherols.

Author Response

Point 1:  ntroduction, line 46-47: it would be more convenient to present UPR regulators in order of their action in the cell during this response: first PERK, then ATF6, and IRE1.

Response 1: The sentence is modified as requested. The UPR regulators are now presented in order of their action in the cell during this response (PERK, ATF6, IRE1).

Point 2: Introduction, line 59, this part of the sentence “T3s are vitamins E with different action and function in respect to  tocopherols” is not very clear. Please, add more information about tocotrienols and tocopherols.

Response 2: The sentence has been rephrased as follows: “T3s, belonging to the vitamin E family of molecules, are often considered of minor action compared to tocopherols [9,10], but have relevant functions related to cell-type dependent growth arrest and apoptosis [10-12]”.

This manuscript is a resubmission of an earlier submission. The following is a list of the peer review reports and author responses from that submission.

Round 1

Reviewer 1 Report

molecules-945432

In this manuscript Ambra et al. analyzed the expression of a subset of miRNAs upon T3 treatment of HeLa cells. They identified several miRNAs and selected miR-190b as a candidate by using in silico evaluation with AGO. By introducing either premiR-190b or antisense miR190b, they claim that modulate XBP1 splicing in HeLa cells treated with tocotrienols.

Although the results the authors present here are potentially interesting, they are too preliminary to be accepted by molecules journal communication. I addition, the manuscript has serious flaws in writing and presentation that make the reviewer impossible to understand the contents. I strongly recommend to check the manuscript extensively and try to point out figures and panels appropriately in the text. Otherwise it is almost impossible to read and understand the findings that the authors have properly.

Major points:

1) It is quite difficult to follow the sentences to describe Figures. For example, in Figure 2, I do not find description which band corresponds to which product. By checking Materials and Methods section, I hardly have a guess for it, but it is not fair to the readers. For Figure 3, the authors describe panel 2 in the sentence, but which is panel 2? How can the readers have an idea for it?

2) The authors used antisense of miR-190b. What is the expected effect with it? Why the behavior of an190b is different in gT3 and dT3 in Figure 4?

3) What is the mechanism for miR-190b to modulate XBP alternative splicing? The authors should discuss it from the point of XBP splicing factor view.

Minor points:

I guess there are some typos and errors in the manuscript (e.g. Figure 5b in page 5 line 143, in page 8 line 229).

Reviewer 2 Report

I wish the authors well during this difficult time, and I congratulate them on a good submission!

The introduction is well written and clear.

I am not particularly happy with the use of a single cell line rather than a panel, but I will differ to the editorial team to decide whether they allow this in this journal or not.

I am happy with the methods used to select specific miRNAs for further analysis based on the Cards results, and the use of Taqman probes for independent validation with RNUB6 as the most frequently used housekeeper in this setting – these methods are well established within the field.

I had not heard of MiREN before. Is it still available? I found the link to it via Leoni & Tramontano’s 2016 paper but the link is dead. I couldn’t find it via Google either – hopefully you can point me in the right direction. Has MiREN only been published once? If the software is very niche, and isn’t available to others, then your data can’t be independently validated or reproduced, which causes me some concern. Is there another more widely available in silico method that you could use to validate the MiREN findings perhaps?

I find it strange that the supplementary figures have been included interspersed within the main text – if this is journal policy it’s fine, I just wanted to mention it in case it’s accidental.

Figure 3 is hard to see - maybe it’s just my monitor, but I imagine other readers will have the same issue if it is. Could you adjust the contrast or apply artificial colour or masking (equally across all panels) in order to make the cells and their morphology more visible? I also note that some of the images within this panel seem to have been taken with different settings, with the middle image brighter for example, meaning we’re not comparing like with like – which settings were fixed on the EVOS and which were adjusted?

Line 144 please amend the phrasing ‘we observed a strong death cells’